# Comparison of Commercial ELISA Kits, a Prototype Multiplex Electrochemoluminescent Assay, and a Multiplex Bead-Based Immunoassay for Detecting a Urine-Based Bladder-Cancer-Associated Diagnostic Signature

**DOI:** 10.3390/diagnostics9040166

**Published:** 2019-10-29

**Authors:** Hideki Furuya, Ian Pagano, Keanu Chee, Takashi Kobayashi, Regan S. Wong, Riko Lee, Charles J. Rosser

**Affiliations:** 1Division of Urology, Cedars-Sinai Medical Center, Los Angeles, CA 90048, USA; Hideki.Furuya@cshs.org (H.F.); rwongega@ucla.edu (R.S.W.); 2Cancer Prevention in the Pacific Program University of Hawaii Cancer Center, Honolulu, HI 96813, USA; pagano@hawaii.edu; 3Translational and Clinical Research Program University of Hawaii Cancer Center, Honolulu, HI 96813, USA; kchee13@punahou.edu (K.C.); riholee@hawaii.edu (R.L.); 4Department of Urology, Kyoto University, Kyoto 606-8507, Japan; selecao@kuhp.kyoto-u.ac.jp

**Keywords:** biomarkers, bladder cancer, multiplex, protein, urine

## Abstract

The ability to accurately measure multiple proteins simultaneously in a single assay has the potential to markedly improve the efficiency of clinical tests composed of multiple biomarkers. We investigated the diagnostic accuracy of the two multiplex protein array platforms for detecting a bladder-cancer-associated diagnostic signature in samples from a cohort of 80 subjects (40 with bladder cancer). Banked urine samples collected from Kyoto and Nara Universities were compared to histologically determined bladder cancer. The concentrations of the 10 proteins (A1AT; apolipoprotein E—APOE; angiogenin—ANG; carbonic anhydrase 9—CA9; interleukin 8—IL-8; matrix metalloproteinase 9—MMP-9; matrix metalloproteinase 10—MMP10; plasminogen activator inhibitor 1—PAI-1; syndecan—SDC1; and vascular endothelial growth factor—VEGF) were monitored using two prototype multiplex array platforms and an enzyme-linked immunosorbent assay (ELISA) according to the manufacturer’s technical specifications. The range for detecting each biomarker was improved in the multiplex assays, even though the lower limit of quantification (LLOQ) was typically lower in the commercial ELISA kits. The area under the receiver operating characteristics (AUROC) of the prototype multiplex assays was reported to be 0.97 for the multiplex bead-based immunoassay (MBA) and 0.86 for the multiplex electrochemoluminescent assay (MEA). The sensitivities and specificities for MBA were 0.93 and 0.95, respectively, and for MEA were 0.85 and 0.80, respectively. Accuracy, positive predictive values (PPV), and negative predictive values (NPV) for MBA were 0.94, 0.95, and 0.93, respectively, and for MEA were 0.83, 0.81, and 0.84, respectively. Based on these encouraging preliminary data, we believe that a multiplex protein array is a viable platform that can be utilized as an efficient and highly accurate tool to quantitate multiple proteins within biologic specimens.

## 1. Introduction

While there continues to be increasing reports of biomarker discovery and early validation studies being published, few new biomarkers have entered clinical practice over the past 30 years. There are many reasons for this, but one of the overwhelming reasons has been the reliance on single biomarkers for the evaluation of cancers that we know can have a broad range of molecular changes. When coupling the tumor’s heterogeneity with the observed variation, even between individuals with similar tumors, it is no wonder very few accurate biomarkers have progressed to clinical relevance. Thus, a shift has occurred to molecular signatures comprised of multiple biomarkers being favored over single biomarkers. The advent of advanced molecular profiling techniques has enabled the derivation of molecular signatures that can more accurately diagnose cancer and make individualized patient evaluation and care feasible. Several molecular signature assays (e.g., Oncotype DX breast, Oncotype DX prostate, and MammaPrint) are now being incorporated into clinical practice [1,2,3]. However, these assays, which are employed to monitor multiple biomarkers, are quite complex and expensive. Thus, these are only offered at centralized laboratories. Technologies that can more easily, efficiently, and simultaneously monitor molecular signatures would be critical in moving robust biomarker panels towards clinical utility.

The objective of the current pilot study was to test the ability of two customized prototype multiplex array platforms to accurately and simultaneously monitor 10 urinary protein biomarkers that comprise our bladder-cancer-associated diagnostic signature; A1AT, APOE, ANG, CA9, IL8, MMP9, MMP10, PAI1, SDC1, and VEGF [4]. The levels of the 10 biomarkers were quantitated in real-world urine samples with 10 commercial enzyme-linked immunosorbent assay (ELISA) kits and two multiplex arrays platforms, then the diagnostic accuracy of the two multiplex array platforms were compared. It is important to note that the multiplex array platforms (prototype) achieved comparable detection ranges as seen with the commercial ELISA kits. Furthermore, both multiplex array platforms were associated with high diagnostic accuracies compared to other urine-based bladder cancer detection assays. We believe this study can validate a urine-based bladder-cancer-associated diagnostic signature in conjunction with the multiplex array platforms.

## 2. Results

### 2.1. Initial Performance of Commercial ELISA Assays and Two Multiplex Array Platforms

Table 1 illustrates the lower level of quantification (LLOQ), upper level of quantification (ULOQ), and coefficient of variation (CV) in a pilot study in which 10 commercial ELISA assays and the two multiplex array platforms (multiplex electrochemoluminescent assay (MEA) and multiplex bead-based immunoassay (MBA)) were assessed in cohort #1. The range for detecting analytes was improved in the multiplex assays, even though the LLOQ was typically lower in the commercial ELISA kits. Except for SDC1 in MEA and A1AT and SDC1 in MBA, CVs were lower in commercial ELISA assays compared to the multiplex array platforms. The data show that the prototype multiplex array platforms perform similarly to commercial ELISA assays, which we extensively tested in this scenario [5,6,7,8].

### 2.2. Initial Diagnostic Accuracy Performance of Multiplex Array Platforms (MEA and MBA)

In order to test the diagnostic accuracy of the two multiplex array platforms, 80 urine samples (cohort #2) were monitored for the 10 biomarkers, using both MEA and MBA. The subjects’ demographics and disease characteristics are summarized in Table 2. Urinary concentrations of 7 of the 10 biomarkers and 3 of the 10 biomarkers were elevated in patients with bladder cancer compared to controls in MEA and MBA, respectively (Table 3). Individual analysis of the ten biomarkers using optimal cutoff values defined by Youden index calculations resulted in the reported area under receiver operating characteristic (ROC) curve (AUROC) (Table 4). AUROC was significantly improved in MBA compared to the MEA for PAI1 (85% vs. 67%; 95% CI: 0.06–0.30; *p* = 0.004) and CA9 (80% vs. 53%; 95% CI: 0.11–0.43; *p* = 0.001). Furthermore, the combination of all ten biomarkers using optimal cutoff values defined by Youden index calculations resulted in an AUROC (Table 4 and Figure 1) of 97% for MBA and 86% for MEA (95% CI: 0.04–0.19; *p* = 0.003). The sensitivities and specificities for MBA were 0.93 and 0.95, respectively, and for MEA were 0.85 and 0.80. Accuracy, positive predictive values (PPV), and negative predictive values (NPV) for MBA were 0.94, 0.95, and 0.93, respectively, and for MEA were 0.83, 0.81, and 0.84, respectively.

## 3. Discussion

There is a growing demand to incorporate molecular signatures into single assays in order to obtain favorable assay properties, such as reduced sample volume, decreased processing time, lower cost of analysis, and lower reagent consumption. Several multiplex protein measurement services are already available (e.g., Meso Scale Discovery, Luminex, Quansys Biosciences Inc, Aushon Biosystems, and RayBiotech, Inc.), and several studies have reported that multiplex ELISA procedures appear suitable and reliable for tissue lysate and serum analyses. The overall approach and goals of protein multiplexing are straightforward—a rapid, cost effective, and reliable detection assay.

Previously, in order to identify a novel, accurate bladder-cancer-associated diagnostic signature, we applied two complementary techniques: gene expression array of shed urothelia in voided urine [9,10] and glycoproteomics of supernatant component of voided urine [11,12]. Then, with subsequent integration of the genomics and proteomics datasets and application of a novel selection algorithm, we derived a bladder cancer diagnostic signature. The signature was confirmed with commercial ELISA kits directed at each of the 10 biomarkers of the signature in a small cohort (*n* = 127; 64 with bladder cancer) [13] and subsequently validated in three large cohorts [4,14,15]. Finally, a meta-analysis was performed to re-evaluate and demonstrate the robustness and consistency of the diagnostic utility of the 10-plex urine-based diagnostic assay. Data pooled from 1173 patients were analyzed. On average, the log odds ratio (OR) for each biomarker was improved by 1.5 or greater when the combination of the 10 biomarkers were used compared to the single biomarker. Furthermore, the combination of the 10 biomarkers showed a higher log OR (log OR: 3.46, 95% CI: 2.60–4.31) than did any single biomarker controlling for grade and stage, thus confirming the utility of our multiplex diagnostic signature [16]. Therefore, our phased, methodical biomarker assay development has identified and validated a urine-based bladder-cancer-associated diagnostic signature.

In the current study, we reported the lower and upper limits of quantification and coefficient of variation between commercial ELISA kits and two multiplex array platforms (MEA and MBA), in addition to reporting the diagnostic performance of the two multiplex array platforms in detecting bladder cancer diagnostic signatures. Firstly, we must stress that all assays are for research use only and have not been optimized for urine sampling or for clinical purposes. The multiplex assays could quantify biomarkers over a wider range than commercial ELISA kits, but commercial ELISA kits typically had a lower level of quantification. Variability, as evidenced by CV, was present in all assays. Briefly, the variability of the ELISA and other methods of chemical assays that produce continuous-type values is summarized by CV, which is defined as the SD divided by the mean, with the result often reported as a percentage [17]. The main appeal of the CV is that the SDs of such assays generally increase or decrease proportionally as the mean increases or decreases, so that division by the mean removes it as a factor in the variability. Except for SDC1 in MEA and A1AT and SDC1 in MBA, CVs were lower in commercial ELISA assays compared to the multiplex array platforms. With further assay development and optimization, CV values should decrease to clinically acceptable levels.

The most encouraging aspect of the current study is the diagnostic accuracy of the multiplex array platforms. Urinary PAI1 was the most accurate single biomarker in both MEA and MBA, with an AUROC of 0.67 in MEA and 0.85 in MBA (95% CI 0.06–0.30). Next, in the MBA, CA9 was the most accurate, with an AUROC of 0.80, while IL8 with an AUROC of 0.60 was the next most accurate biomarker in the MEA. Thus, the difference between the MBA and MEA assays were due to the ability of the platforms to more accurately detect concentration of these analtyes. As we previously reported, when all 10 biomarkers are combined we see the best accuracy, with an AUROC of 0.86 in MEA and 0.97 in MBA. Thus, the bladder cancer signature comprised of the 10 biomarkers achieved greater diagnostic power than any biomarker alone and has the potential to consider the heterogeneity of bladder tumors.

Though the sensitivity and specificity achieved by the described multiplex array platforms far exceed those achieved by voided urine cytology (32% sensitivity and 94% specificity) or single biomarker tests for bladder cancer detection [18], we feel that with assay refinement and optimization, the sensitivity and specificity of these multiplex assays can only improve even further.

Clinically, accurate non-invasive bladder cancer tests would have a clear impact on the clinical management of patients with bladder cancer. Therefore, our goal is to be able to detect bladder cancer in a swifter and more expedient manner (i.e., when it is in its earlier stages), so that the patient can expect an improved survival and perhaps improved quality of life. Therefore, to successfully translate a diagnostic test from the basic science laboratory to the clinic, the diagnostic needs to be cost-effective, as well as accurate, especially if it is to be used over a long period of follow-up, as in the case for bladder cancer. The detection of urinary proteins through multiplex array platforms has the potential to be relatively simple to perform, interpret, and is affordable enough to meet this requirement. With that said, the multiplex bead-based immunoassay (MBA) is currently being applied to quantitatively detect and monitor a bladder-cancer-associated diagnostic signature in three large multicenter, international prospective clinical trials (NCT 03193515, 03193528, and 03193541).

## 4. Materials and Methods

### 4.1. Patients and Specimen Processing

Ethical review of the study was performed by Western Institutional Review Board. Banked voided urine samples were made available for analysis. Urine samples were collected prior to any instrumentation, centrifuged to remove cellular pellets, and stored frozen until analysis, as previously reported [4,5,6,7,8,9,10,11,12,13,14,15]. The study was comprised of cohort #1 (*n* = 34; 6 with bladder cancer) from University of Hawaii Cancer Center (UHCC) to assess the lower level of quantification (LLOQ), upper level of quantification (ULOQ), and coefficient of variation (CV); and cohort #2 (*n* = 80; 40 with bladder cancer) from Kyoto University and Nara Medical University to assess diagnostic accuracy of the multiplex array platforms. For the bladder cancer cases, histological confirmation of urothelial carcinoma, including grade and stage, was defined from excised tissue.

Urine samples were monitored for protein content by using a Pierce 660-nm protein assay kit (Thermo Fisher Scientific Inc., Waltham, MA, USA), while the concentration of urinary creatinine was measured using an enzymatic assay (Cat#KGE005 R&D Systems Inc., Minneapolis, MN, USA) according to the products’ manufacturer instructions. Laboratory personnel were blinded to final diagnoses.

### 4.2. Commercial Enzyme-Linked Immunosorbent Assays (ELISA)

Levels of A1AT (Cat# ab108799, Abcam, Cambridge, MA, USA), apolipoprotein E (APOE, Cat # KA 1031 Abnova, Walnut, CA, USA), angiogenin (ANG, Cat# CK400 CellSciences, Canton, MA, USA), carbonic anhydrase 9 (CA9, Cat# DCA900 R&D Systems Inc., Minneapolis, MN, USA), interleukin 8 (IL-8, Cat # ab46032 Abcam, Cambridge, MA, USA), matrix metalloproteinase 9 (MMP-9, Cat# DMP900 R&D Systems Inc., Minneapolis, MN, USA), matrix metalloproteinase 10 (MMP-10, Cat# DMP1000 R&D Systems Inc., Minneapolis, MN, USA), plasminogen activator inhibitor 1 (PAI-1, Cat# EA-0207 Signosis Inc., Sunnyvale, CA, USA), syndecan (Cat# ab46507 Abcam, Cambridge, MA, USA), and vascular endothelial growth factor (VEGF, Cat # 100663 Abcam, Cambridge, MA, USA) were monitored in voided urine samples using commercial enzyme-linked immunosorbent assays (ELISA), as previously reported [4,5,6,7,8]. These assays were performed as per manufacturer’s instructions. Standard curves were prepared using purified standards for each protein assessed. Curve fitting was accomplished by four-parameter logistic regression, following manufacturer’s instructions.

### 4.3. Multiplex Array Platforms

The concentrations of the 10 proteins (A1AT, APOE, ANG, CA9, IL8, MMP9, MMP10, PAI1, SDC1, and VEGF) were monitored using two prototype multiplex array platforms; multiplex electrochemoluminescent assay (MEA) from Meso Scale Discovery (Rockville, MD, USA) and multiplex bead-based assay (MBA) from R&D Systems Inc. (Minneapolis, MN, USA) for Luminex 200. Final monoclonal antibody pairs (capture and detection) for the multiplex assays were selected based on sensitivity, specificity, physical properties, and recognition of native protein, as described previously [19]. In order to ensure detection across the range of protein concentrations with negligible interference and cross-reactivity, in the MEA 7 of the 10 analytes (IL8, MMP9, MMP10, APOE, PAI1, CA9, and VEGFA) were multiplexed in one well and the other 3 assays (ANG, SDC1, A1AT) in a separate well, while in the MBA 9 of the 10 analytes (IL8, MMP9, MMP10, APOE, PAI1, CA9, VEGFA, ANG, SDC1) were multiplexed in one well and one analyte (A1AT) in a separate well. Urine samples were diluted 4-fold for the 7-plex assay and 200-fold for the 3-plex MEA, while urine samples were diluted 2-fold for the 9-plex assay and 5-fold for the 1-plex MBA. A five-point standard curve across the 4-log dynamic range of the assays was included in the current assay design. Urine samples were handled on ice and diluted with appropriate assay diluent. Samples and standards (50μL) were added to the 96-well plates in triplicate and analyzed as per assay protocol. MEA plates were read on the QuickPlex^®^ SQ 120 (MSD) instrument, while the MBA plates were read on the Luminex 200. As per above, calibration curves were generated along with optimal fit in conjunction with Akaike’s information criteria (AIC) values [19,20].

### 4.4. Data Analysis

Descriptive statistics were used to report the LLOQ and ULOQ, as well as CV in cohort #1. In cohort #2, Wilcoxon rank sum tests were used to determine the association between each biomarker and bladder cancer. We investigated the diagnostic performance of the protein biomarkers for bladder cancer (BCa) detection using the logistic regression analysis with BCa status (yes vs. no) as the response variable and 10 biomarkers as the explanatory variables. The individual biomarkers were combined into a linear combination, with the regression coefficients obtained in logistic regression as the weights, and the linear combination used as a combined score for the detection of BCa. Using cutoff thresholds previously reported [4,5,6,7,8,9,10,11,12,13,14,15,19,20], we defined a diagnostic test that is either positive or negative when the linear combination of biomarkers is either ≥ or < the cutoff. Then, for a given cutoff threshold, we calculated the sensitivity and specificity of the test. We generated a ROC curve by plotting values for sensitivity against the false-positive rates (1—specificity) for various cutoff thresholds [21]. The relative ability of the combination of biomarkers to indicate BCa was estimated by calculating the area under the ROC curves (AUC), with a higher AUC indicating a stronger predictor. We selected the optimal cutoff value defined by the Youden index [22] (i.e., the cutoff value that maximizes the sum of the sensitivity and the specificity). We estimated the sensitivity, specificity, positive prediction value (PPV), and negative prediction value (NPV) of the combination of biomarkers at the optimal cutoff value. Statistical significance in this study was set at *p* < 0.05 and all reported *p* values were 2-sided. All analyses were performed using SAS software version 9.3 (SAS Institute Inc., Cary, NC, USA).

## 5. Conclusions

Based on these encouraging early results, we believe that the multiplex array platform is a feasible avenue to exploit as a simple, yet accurate tool to quantitate the amount of 10 diagnostic proteins in voided urine samples. At this stage in assay development, the detection range, variability, and accuracy are acceptable, and thus the method warrants further development. Furthermore, the prescribed platform is already in many clinical laboratories, thus facilitating clinical uptake of a new robust assay.

## Figures and Tables

**Figure 1 diagnostics-09-00166-f001:**
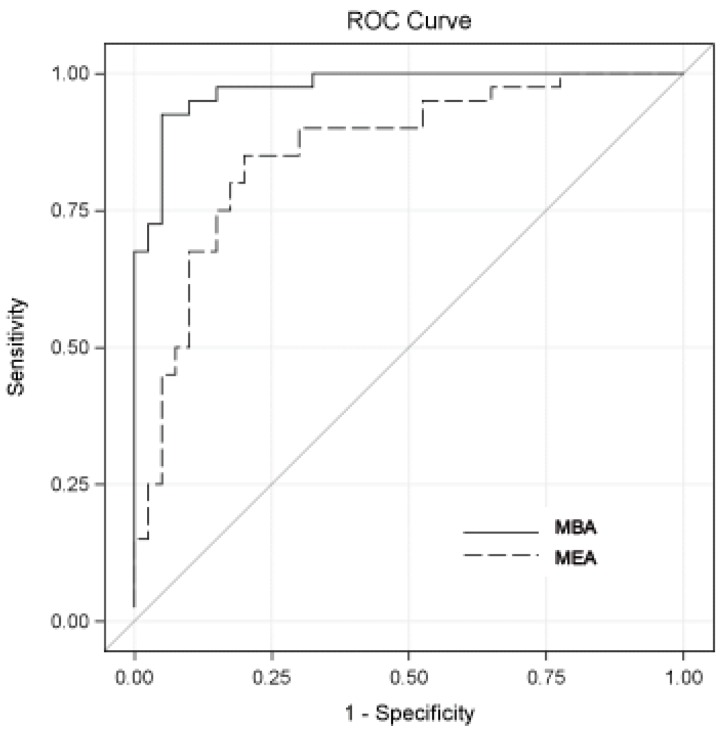
Diagnostic performance of bladder-cancer-associated molecular panels combining all biomarkers. A receiver operating characteristic (ROC) curves for MBA (solid line) and MEA (dashed line) assays are shown. The areas under the curves were 0.97 for MBA and 0.86 for MEA. The sensitivity and specificity values maximizing the Youden index (sensitivity ± specificity − 1) for MBA were 0.93 and 0.95, respectively, and for MEA were 0.85 and 0.80. Accuracy, positive predictive values (PPV), and negative predictive values (NPV) for MBA were 0.94, 0.95, and 0.93, respectively, and for MEA were 0.83, 0.81, and 0.84.

**Table 1 diagnostics-09-00166-t001:** Lower limits of quantification, upper limits of quantification, and coefficient of variation between commercial ELISA kits and two multiplex array platforms for detecting 10 urine-based biomarkers associated with bladder cancer.

	Commercial ELISA Kits			MEA			MBA		
Analyte	CV	LLOQ (pg/mL)	ULOQ (pg/mL)	CV	LLOQ (pg/mL)	ULOQ (pg/mL)	CV	LLOQ (pg/mL)	ULOQ (pg/mL)
IL8	2.85	0.5	2033	3.65	1.23	3169	3.58	2.01	1397
MMP9	1.69	0.1	9.0	3.14	23.9	9337	3.37	14.76	7164
MMP10	0.00	4.13	4.13	3.01	8.94	816	0.77	1.71	90.33
VEGFA	0.91	10.0	506	1.52	21.38	2587	1.61	18.24	1031
CA9	0.00	2.28	2.28	1.47	2.06	132.32	1.04	0.48	54.07
APOE	0.31	0.03	0.087	4.32	435.81	653309	2.28	78.30	1998
PAI1	0.38	0.05	0.168	4.46	4.47	73437	2.86	0.10	426.92
A1AT	2.63	6.0	5968	3.13	4422	682690	1.55	424.31	179,724
ANG	1.74	2.9	2208	2.33	18.35	19638	2.27	24.98	6557
SDC1	1.16	9.88	474	0.84	1002	38916	0.60	2608	19,051

Abbreviations: CV = coefficient of variation; LLOQ—lower level of quantification; ULOQ—upper level of quantification; MEA—multiplex electrochemoluminescent assay; MBA—multiplex bead-based immunoassay; apolipoprotein E—APOE; angiogenin—ANG; carbonic anhydrase 9—CA9; interleukin 8—IL-8; matrix metalloproteinase 9—MMP-9; matrix metalloproteinase 10—MMP10; plasminogen activator inhibitor 1—PAI-1; syndecan—SDC1; and vascular endothelial growth factor—VEGF.

**Table 2 diagnostics-09-00166-t002:** Demographic and clinical-pathologic characteristics.

	Cancer Cases (*n* = 40)	Controls (*n* = 40)	*p*
Age			
30–65	6 (15%)	12 (30%)	0.08
66–70	7 (18%)	12 (30%)	
71–75	13 (33%)	10 (25%)	
76–90	14 (35%)	6 (15%)	
Sex			
Male	36 (90%)	30 (75%)	0.14
Female	4 (10%)	10 (25%)	
Cytology			
Negative	19 (48%)		
Positive	13 (33%)		
N/A	8 (20%)	40 (100%)	
Grade			
Low	12 (30%)		
High	21 (53%)		
N/A	7 (18%)	40 (100%)	
Stage			
Tis	1 (3%)		
Ta	21 (53%)		
T1	4 (10%)		
T2	6 (15%)		
N/A	8 (20%)	40 (100%)	

Note. The *p*-value is for Fisher’s exact test.

**Table 3 diagnostics-09-00166-t003:** Geometric mean (± SD) urinary concentrations of biomarkers assessed by MEA and MBA.

Biomarker	MEA		MBA			
	Cancers	Controls	*p*	Cancers	Controls	*p*
MMP9 (pg/mL)	749.0 ± 5949.9	439.5 ± 4964.8	0.31	13.2 ± 355.5	36.2 ± 388.8	0.13
IL8 (pg/mL)	13.9 ± 189.4	5.8 ± 21.7	0.08	8.0 ± 66.9	5.1 ± 19.5	0.31
VEGF (pg/mL)	205.4 ± 606.8	207.2 ± 380.3	0.97	38.3 ± 121.8	41.8 ± 79.0	0.76
SCD1 (ng/mL)	4.9 ± 22.9	4.7 ± 7.6	0.93	5.9 ± 4.8	6.2 ± 21.5	0.84
ANG(pg/mL)	140.7 ± 714.8	163.2 ± 488.2	0.68	69.1 ± 321.8	93.1 ± 446.9	0.45
PA1 (pg/mL)	32.1 ± 225.9	9.9 ± 19.5	0.002	4.4 ± 28.1	1.3 ± 1.3	0.001
CA9 (pg/mL)	7.7 ± 3 4.6	5.3 ± 12.3	0.26	11.2 ± 22.0	2.7 ± 6.6	<.0001
MMP10 (pg/mL)	15.0 ± 91.9	6.5 ± 44.3	0.07	2.3 ± 9.4	4.8 ± 24.7	0.06
A1AT (ng/mL)	18.5 ± 484.0	7.1 ± 170.0	0.19	20.3 ± 82.2	25.6 ± 97.6	0.52
APOE (ng/mL)	7.1 ± 18.3	2.3 ± 44.6	0.03	1.1 ± 2.4	1.2 ± 2.6	0.55

Note. There are 40 cancer cases (creatinine 71.6 ± 54.3 mg/dl) and 40 controls (creatinine 93.1 ± 91.7 mg/dL).

**Table 4 diagnostics-09-00166-t004:** Assay area under the ROC curve (AUROC) comparisons of MEA and MBA for bladder cancer diagnostic biomarkers.

Biomarker	MEA	MBA	Difference	95% Confidence Interval	*p*
MMP9	0.48	0.64	0.16	(−0.08, 0.39)	0.19
IL8	0.60	0.53	−0.07	(−0.16, 0.03)	0.15
VEGF	0.54	0.58	0.04	(−0.07, 0.16)	0.47
SDC1	0.57	0.62	0.05	(−0.17, 0.27)	0.65
ANG	0.53	0.56	0.04	(−0.08, 0.15)	0.55
PA1	0.67	0.85	0.18	(0.06, 0.30)	0.004
CA9	0.53	0.80	0.27	(0.11, 0.43)	0.001
MMP10	0.59	0.62	0.03	(−0.15, 0.21)	0.75
A1AT	0.57	0.55	−0.02	(−0.24, 0.19)	0.82
APOE	0.48	0.59	0.11	(−0.11, 0.33)	0.32
Combined	0.86	0.97	0.12	(0.04, 0.19)	0.003

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
