# Peer review of "Comparison of Commercial ELISA Kits, a Prototype Multiplex Electrochemoluminescent Assay, and a Multiplex Bead-Based Immunoassay for Detecting a Urine-Based Bladder-Cancer-Associated Diagnostic Signature"

_diagnostics, 2019, doi:10.3390/diagnostics9040166_

Round 1
Reviewer 1 Report
The title of the article “Comparison of commercial ELISA kits and prototype multiplex electrochemoluminescent assay and multiplex bead-based immunoassay to detect a urine based bladder cancer-associated diagnostic signature”
The article is indeed written very well taking into consideration of the current need for multiple biomarkers identification assays in single platform assays. The data is analyzed and presented in accurate way using statistical approaches, appreciable. Any specific reason the author selected 10 specific biomarkers in molecular signature assays? The author need to explain in Table 2: What kind of cytology they did in patient’s samples and not done in control.Author Response
Any specific reason the author selected 10 specific biomarkers in molecular signature assays?
Over the past 13 years we have undertaken a phase approached to identify and validate a urine-based diagnostic signature. The phase approach included gene expression array followed by RT-PCR validation in an independent cohort, along with glycoproteomics (LC-MS/MS) followed by WB/ELISA validation in an independent cohort. Then a selection algorithm was used to identify a signature which was narrowed down to these 10 biomarkers (APOE, ANG, A1AT, CA9, IL8, MMP9, MMP10, PAI1, SDC1, VEGF). These 10 biomarkers were subsequently validated in 5 large cohorts totaling over 1700 patients. Based on these results we designed the Luminex multiplex kit which is reported in the current manuscript. This is noted in the references to the following references (12-15, 19-23).
The author need to explain in Table 2: What kind of cytology they did in patient’s samples and not done in control.
Voiding urinary cytologies were obtain in the group with bladder cancer. None of the controls had voided urinary cytology.
Reviewer 2 Report
This a very wellwritten paper and an interesting topic but several aspects need to revised.
Urine samples collected from Kyoto and Nara Universities were compared. Where cases and controls distributed equally among both recruitment centres? Please provide more info The abbreviations make difficult to follow the paper, especially when they are specified in the last section(methods). Both cohorts are independent? Authors should analyse results from table 1 considering separately normal samples and cancer cases samples. By the way only 6 cases were included, not a very large numbers. Could they include more cases and give the results separately from the controls? how do they explain differences in the markers when comparing values from mea and mba (table 3) or even table 4? Authors should provideonfidence intervals and pvalues for the AUC values Were the values of the AUc difference when not considering the proteins that were significant different among both multiplex array platform? It is unclear how they obtain the Roc curves. I suppose that first they have the values of each of the proteins in the platform and then?? How do they obtain the classification? In how many cases the result of the proteins were under or below ULOQ or LLOQ? Some words need to be revised, for example line 118Author Response
Where cases and controls distributed equally among both recruitment centres?
The cases and controls were not distributed equally from both centers.
Please provide more info The abbreviations make difficult to follow the paper, especially when they are specified in the last section(methods).
Done
Both cohorts are independent?
The study is comprised of 2 cohorts. Cohort #1 has 34 subjects (6 with cancer) from UHCC and Cohort #2 has 80 subjects (40 with cancer). Both cohorts were run on the 2 platforms.
Authors should analyse results from table 1 considering separately normal samples and cancer cases samples. By the way only 6 cases were included, not a very large numbers. Could they include more cases and give the results separately from the controls?
The goal of this project was not to confirm our bladder cancer associated diagnostic signature. This has been done in previous manuscripts. The goal of this project was not even to report on the ability of a multiplex assay to detect this urine-based bladder cancer associated signature. The goal of the project was to compare 2 multiplex assays on different platforms. Though it would be ideal to have a large cohort comprised of well balanced groups, what we need here is to run samples and report key features as it relates to analytical testing of the two assays. This would include the LLOQ (lower level of quantification), ULOQ (upper level of quantification) and CV (coefficient of variation). In the manuscript, we have clearly illustrated that the multiplex bead based immunoassay has similar characteristic as our well reported multiplex electrochemoluminescent assay. Now with this foundation we will set out in subsequent studies for the clinical validation.
how do they explain differences in the markers when comparing values from mea and mba (table 3) or even table 4?
These are 2 different assays using different paired antibodies to detect these analytes in the urine.
Authors should provide confidence intervals and p values for the AUC values
Done.
Were the values of the AUc difference when not considering the proteins that were significant different among both multiplex array platform?
When considering the AUC curves that were not significantly different between the 2 groups was the AUC different. Not they were not different. But by adding the remaining analtyes we were able to better protect the algorithm from error.
It is unclear how they obtain the Roc curves. I suppose that first they have the values of each of the proteins in the platform and then? How do they obtain the classification? In how many cases the result of the proteins were under or below ULOQ or LLOQ?
We have extensively published on this and have added the references.
Some words need to be revised, for example line 118
Done
Round 2
Reviewer 2 Report
For the reader it will be better if the statistical section is more detailed.
The differences among both results is just because there are some markers that differ. this might be highlighted in the text.
Author Response
For the reader it will be better if the statistical section is more detailed.
A more detailed stats section was added.
The differences among both results is just because there are some markers that differ. this might be highlighted in the text.
A sentence was added to the discussion to highlight this aspect.